# Evaluation of the Nutritive Value and Digestibility of Sprouted Barley as Feed for Growing Lambs: In Vivo and In Vitro Studies

**DOI:** 10.3390/ani12091206

**Published:** 2022-05-07

**Authors:** Hani H. Al-Baadani, Abdullah N. Alowaimer, Mohammed A. Al-Badwi, Mutassim M. Abdelrahman, Walid H. Soufan, Ibrahim A. Alhidary

**Affiliations:** 1Department of Animal Production, College of Food and Agriculture Science, King Saud University, P.O. Box 2460, Riyadh 11451, Saudi Arabia; hsaeed@ksu.edu.sa (H.H.A.-B.); aowaimer@ksu.edu.sa (A.N.A.); malbadwi@ksu.deu.sa (M.A.A.-B.); amutassim@ksu.edu.sa (M.M.A.); 2Department of Plant Production, College of Food and Agriculture Science, King Saud University, P.O. Box 2460, Riyadh 11451, Saudi Arabia; wsoufan@ksu.edu.sa

**Keywords:** growing lambs, nutrient digestibility, performance, rumen fermentation, sprouted barley

## Abstract

**Simple Summary:**

Sprouted barley has recently been used in animal feed, but studies on its nutritional and digestibility values for growing lambs are very limited, especially when compared with traditional forages. However, the aim of the current research is to determine the practical level and nutritional properties of sprouted barley as a replacement for the traditional feed for growing lambs by ascertaining its efficacy of performance, nutritional value, digestibility, and fermentation characteristics in vivo and in vitro. The current research helps both workers and researchers to determine the level and the most effective way to utilize sprouted barley as an alternative strategy to traditional feeding systems. In conclusion, sprouted barley with traditional feed improves digestibility and fermentation characteristics. Further studies are needed to increase nutrient requirements for optimal lamb growth performance.

**Abstract:**

The main objective of this study was to investigate the effects of freshly sprouted barley on the growth of lambs, in addition to its nutritional value and digestibility. In addition, sprouted barley digestibility and rumen fermentation were studied in vitro on a dry matter (DM) basis. A total of 45 three-month-old Awassi lambs were randomly assigned to five treatments of sprouted barley (0, 25, 50, 75, 100%) diets. Bodyweight, weight gain, feed intake and feed efficiency were recorded every two weeks. Nutrient analyses were performed on feed, faecal, and urine samples. DM and non-fibrous carbohydrates were measured. Digestibility of DM, organic matter (OM), and neutral detergent fiber (NDF), as well as gas production, pH value, ammonia-N, and volatile fatty acids (VFAs), were determined in vitro using continuous culture. The results showed that final bodyweight was lower (*p* < 0.05), while feed intake and the feed-to-gain ratio were increased (*p* < 0.05) in sprouted barley treatments. Nutrient analysis indicators of sprouted barley treatments (25 to100%) were lower (*p* < 0.05) for DM, crude protein, acid detergent fiber, lignin and ash, and higher for total digestible nutrients, NDF, fat, phosphorus, zinc, copper, and net energy than the traditional diet. In the in vivo study, the digestibility of nutrients in sprouted barley treatments was improved (*p* < 0.05), while the diet (sprouted barley 100%) had the lowest digestibility of DM, OM, and NDF compared with the other treatments in the in vitro study. In conclusion, the addition of sprouted barley improved digestibility, and fermentation characteristics, while having a negative effect on growth. Further studies are recommended for optimal growth performance.

## 1. Introduction

Lambs are an important resource that support food security in many countries, which are mostly raised on extensive grazing systems. These systems differ in terms of quality and quantity, which depend mainly on climatic variations, including temperature and precipitation, and are also seasonal [1]. However, the nutrient requirements of grazing animals are not often met under such systems to allow them to reach their productive efficiency [2]. Due to the lack of rangeland fodder, people have been compelled to switch to alternative feed sources and, as a result, their production patterns have shifted to semi-intensive systems using traditional feeds such as grain and rough fodder. At the same time, there are obstacles that herd owners face in using these systems, including high prices globally and waste from water source consumption [3]. Sprouted barley is a new way of producing feed forages without using soil, with a high germination rate and a fast-growing period. This method could be especially important in regions where water shortages and the seasonality of forages are common challenges for livestock producers [4]. Sprouted grains are efficiently digested compared to grain seeds because of the high activation of hydrolytic enzymes as a result of germination [5]. Consequently, Fazaeli et al. [6] reported that hydrolytic enzymes convert proteins, starch and fat into simple forms of amino acids, sugars, and fatty acids. Furthermore, the sprouted process increases the content of crude fiber [7], chelates of minerals [8], and decreases the content of phytic acid and protease inhibitors, as well as many other anti-nutrients [9]. In addition, the important benefit of producing sprouted barley is the minimal water consumption compared to the conventional production system. Germination has been demonstrated to be an inexpensive (low-cost process) and sustainable process that improves nutrient quality and the content of functional compounds of grains, as well as their palatability, digestibility, and bioavailability [10,11]. However, the magnitude of changes caused by germination depends on the grain variety and germination conditions [12]. Dung et al. [13] indicated that the benefit of sprouted barley in lamb feed may be negated by the total dry matter (DM) loss, with no improvement in nutrient concentrations or digestibility. Several studies suggest that feeding sprouted barley increases performance only in animals that do not receive adequate protein, energy, or minerals [14], or that the readily available nutrients in sprouted barley may stimulate enhanced utilization of poor-quality feed [15]. 

To our knowledge, the nutritional value and digestibility of sprouted barley, especially compared to traditional forages such as barley grain and alfalfa hay, have not been extensively demonstrated in previous studies. The hypothesis of this study is that sprouted barley could be an alternative strategy to traditional feed for growing lambs, with the identification of factors resulting from the substitution of animals. Therefore, the main objective of the current study is to investigate the effects of freshly sprouted barley levels with traditional feed on growth performance, nutritional value, and digestibility in growing lambs. Additionally, to evaluate sprouted barley levels with traditional feed (DM basis) on digestibility, CO_2_ production, and fermentation characteristics in vitro using a continuous culture fermentation.

## 2. Materials and Methods

Animal husbandry and sampling were carried out in accordance with the procedures established by the Scientific Research Ethics Committee, King Saud University, Saudi Arabia (Ethics Reference No: KSU-SE-22-01), also with the care and use of farm animals in research in the United States of America, Animal Science Association [16].

### 2.1. Sprouted Barley Production

The use of hydroponics for the cultivation of sprouted barley was carried out according to the method described by Al-Saadi and Al-Zubiadi [17]. Briefly, the sprouted barley production plan was carried out in a hydroponic steel chamber (3.0 m length × 2.5 m height × 2.5 m width) in the Department of Food and Agriculture Science, King Saud University, Saudi Arabia. The hydroponic chamber was designed to accommodate 140 trays (70 cm length × 30 cm width) with a capacity for seven growth stages (7 days). Each tray had an automated sprinkler watering system, and the conditions inside the chamber were managed in terms of ventilation, heat, and relative humidity by conditioning and circulating the air to maintain a constant temperature range of 18–20 °C and relative humidity of around 75%. Fluorescent lighting of about 1 watt/cm^3^ was used throughout the day in a vertical position for leaf development. Barley seeds were purchased locally. About 2 kg of seeds were placed in each tray after cleaning and washing and soaking in water for 24 h. On the seventh day of the growth stage, when they had reached a height of 18–20 cm, the carpets were removed and aired for 24 h to dry them further before being cut and presented to the animals, with the production cycle continuing daily during the study period.

### 2.2. Diets Sampling and Analysis

During the study period, every 15 days of the study (five replicates), feed samples were collected in the same levels of sprouted barley given to the treated lambs, dried (60 °C) to determine the initial moisture content and then ground to a fine powder. According to the previously described method [2], the forage powder samples were analyzed in triplicate to estimate the content of nutrients such as dry matter (DM) by drying overnight at 105 °C in a drying oven (Sanyo convection oven, Osaka, Japan), crude protein (CP; Kjeldahl method, using an N conversion factor of 6.6), crude fat, ash, total digestible nutrients (TDN), and net energy (NE) according to the methods of the Association of Official Analytical Chemists [18]. According to Van Soest et al. [19], fiber fractions such as neutral detergent fiber (NDF), acid detergent fiber (ADF), and lignin were determined. Organic matter (OM) was calculated as OM % = 100-ash [20]. Non-fibrous carbohydrates (NFC) were calculated using the equation NFC (%) = 100% − (CP + total fiber + crude fat + ash) according to Bachmann et al. [21]. The content of macro and micro minerals such as calcium, phosphorus, magnesium, potassium, sulfur, sodium, zinc, and copper were digested in a mixture of sulfuric acid and hydrogen peroxide (105 °C) in a closed microwave using the method previously described [6,22]. All minerals were determined by an atomic absorption spectrometer (PerkinElmer, instruments, Analyst, Waltham, MA, USA) using the Association of Official Analytical Chemists [18].

### 2.3. Housing Lambs and Experimental Design 

A total of 45 growing Awassi male lambs (27.85 ± 2.5 kg) were used for the present study for 75 days. They were purchased at the age of 3 months from local trustworthy farms, and then brought to the Experimental farm of the Department of Animal Production Department, the University of Food and Agriculture Sciences, King Saud University, Riyadh. Lambs were weighed individually and then randomly divided into 45 individual pens (150 × 120 cm) in five feeding treatments, each pen representing one experimental unit (nine replicates per treatment with one lamb per replicate), based on a completely randomized design under the natural winter environmental conditions of the region. Feed treatments were distributed as follows: T1: 100% added traditional feed (control; 70% barley grain + 30% alfalfa hay), T2: 25% added barley sprouts with 75% traditional feed, T3: 50% added barley sprouts with 50% traditional feed, T4: 75% added barley sprouts with 25% traditional feed and T5: 100% added barley sprouts. The feed ingredients and nutrient composition of all dietary treatments are listed in Table 1. Before the start of the actual study, all lambs were acclimatized to the used diet for 14 days, vaccinated against enterotoxaemia and septicemia, and PPR subcutaneously inoculated with an agent against ecto-/endoparasites according to the recommendations of the Directorate of Animal Resources of the Ministry of Environment, Water and Agriculture [23] in Saudi Arabia, Veterinary Vaccine Centre (manufactured by Ibrize Co. Riyadh, Saudi Arabia). During the study period, all animals were provided ad libitum access to feed and water, as well as up to 5% additional feed daily to reach refusal.

### 2.4. Parameters of Growth Performance 

All lambs were weighed to determine initial body weight and final body weight (1 and 75 days) to determine average weight gain and daily gain (final body weight − initial body weight). Average daily feed intake (amount of feed rejected − feed offered) was also measured to determine the feed-to-gain ratio (feed intake/weight gain) according to Pereira et al. [24]. Relative growth (RG) was calculated using the following equation: RG = 2 (final body weight − initial body weight)/(final body weight + initial body weight) × 100 according to Goiri et al. [25]. 

### 2.5. Digestibility Procedure and Analysis In Vivo

At the end of the 75-day feeding experiment, five lambs were randomly selected from each treatment and housed in metabolism cages (120 × 80 cm) to perform a digestibility experiment according to the methodology described by Omar [26]; Al-Saadi and Al-Zubiadi [17]. The digestion experiment lasted eight days, beginning with a four-day acclimation phase for all animals in the cages, followed by a four-day collection period, during which daily feed intake and fecal and urine excretion were recorded. Feed samples (both offered and rejected), feces, and urine samples were collected and stored at −20 °C until nutrient analysis. Urine was analyzed for nitrogen content, while fecal samples were subjected to the same tests as the feed samples. Apparent digestibility (AD) was calculated on a DM basis using the following formula: AD % = [(intake − fecal excretion)/intake] × 100 for each animal according to Bachmann et al. [21].

### 2.6. Digestibility Procedures and Gas Production of Diets In Vitro 

Diet samples (five dietary treatments; T1:T5) were collected and dried to process DM and stored in sterile bags at 5 °C until apparent digestibility could be estimated using the method of Embaby et al. [27]. In four ANKOM gas jars per treatment as replicates (20 gas jars/treatment), 70 mL of strained rumen fluid was combined with 130 mL of pre-warmed buffer medium (as batch rumen cultures). Then, 200 g of finely powdered feed (based on DM) was weighed into a Dacron bag (Ankom Inc., Fairport, NY, USA) and then added to each of the ANKOM jars, and then incubated for 24 h to determine the apparent digestibility of DM, OM, and NDF. The samplers and feed treatments were analyzed for DM calculation, ash analysis for calculation of OM (100-Ash), and NDF according to the methods of the Association of Official Analytical Chemists [18]. All jars of dietary treatments, which were previously sealed, were filled with CO2 and then connected to a Tedlar gas collection bag (Santa Ana, CA, USA). The jars were placed in a water bath (Thermo Fisher Scientific, model 2873, Waltham, MA, USA) at 39 °C for 24 h until the release process of the gasses in the collection bag was completed. During this process, the jars were shaken for 30 s every 2 h. All CO2 gas production values were expressed as milliliters per time (2 h) and total gas production (24 h) [28].

### 2.7. Measurement of Fermentation Characteristics of Diets In Vitro 

To determine the volatile fatty acids (VFA) and Ammonia-N (NH3-N) in the culture jar at the end of each experiment. From each jar, 5 mL of sample was taken and placed directly in an ice bath and then stored at −20 °C until analysis. Samples were mixed with 1 mL of 25% meta-phosphoric acid and centrifuged at 20,000× *g* for 10 min at 4 °C to produce a clear supernatant. One ml of the supernatant was filtered with a PTFE syringe filter (0.2 μm) and transferred to a 1.5 mL glass chromatography vial (Agilent). Acetic acid (C2), propionic acid (C3), butyric acid (C4), iso-butyric acid (iso-C4), valeric acid (Val), and iso-valeric acid (iso-Val) were analyzed on a gas chromatograph (Shimadzu Scientific Instruments Inc., Columbia, MD, USA) using 2-ethylbutyric acid as an internal standard [29]. Separation of VFAs was performed as previously described [30]. VFAs were expressed as mM/1 mL samples. A sample was taken from each of the jars and centrifuged at 12,000× *g* for 15 min at 4 °C to measure the ammonia N concentration (NH3-N) using a spectrophotometer (Perkin Elmer, Waltham, MA, USA) after being acidified with 0.5 mL of 0.1 N HCl, according to the method of Abdelrahman et al. [31]. The pH values of the culture jar sample were determined directly during sampling using a digital pH meter (Model pH 211; Hanna Instruments, Woonsocket, RI, USA).

### 2.8. Statistical Analysis

For both studies, all data were analyzed as a completely randomized design using general linear model procedures of Statistical Analysis System software [32] based on animal and jar as the experimental unit. Only the use of the collected samples for chemical composition analysis of the diet precluded statistical comparison. Statistical differences (*p* < 0.05) between the means of the dietary treatments were determined using Duncan’s multiple range test and were also analyzed for linear or quadratic responses with orthogonal contrasts (T1 vs. T2 + T3 + T4 + T5). All means were reported with the standard error of the means (means ± SEM). 

## 3. Results

The chemical composition of the dietary treatments on a DM basis is shown in Table 1. DM was lower with an increasing proportion of sprouted barley at 25 to 100% with 75 to 0% traditional feed (T2 to T5) compared to 100% added traditional feeding system (T1; 70% barley grains + 30% alfalfa hay). However, a numerical increase in NDF, fat, phosphorus, zinc, copper, TDN, and NE concentrations of 2.69%, 1.01%, 0.16%, 37.0 ppm, 1.0 ppm, 3.40%, and 0.03 Mcal/lb, respectively, was observed in sprouted barley (T5) compared to the traditional feeding system (T1). CP, ADF, lignin, ash, and some minerals (calcium, magnesium, potassium, sulphur, and sodium) were numerically lower in the sprouted barley than in the traditional feeding system.

The general growth performance of growing lambs fed on dietary treatments with sprouted barley is shown in Table 2. The results of the in vivo study show that the addition of sprouted barley at 25 to 100% with 75 to 0% traditional diet (T2 to T5; respectively) had the lowest (*p* < 0.05) final body weight (FBW) at 75 days of the study period compared to 100% added traditional feeding system (T1; 70% barley grains + 30% alfalfa hay). Bodyweight gain (BWG) and average daily gain (ADG) were lower at T5 (*p* < 0.05), while there were no significant effects at T2, T3 and T4 compared to T1. The comparison (T1 vs. T2 + T3 + T4 + T5) showed a linear effect (*p* < 0.05) in FBW, BWG, and ADG with increasing sprouted barley content. There was a linear and quadratic (*p* < 0.05) increase in average daily intake and feed to gain ratio with increasing sprouted barley content (T2 to T5) compared to T1 (1 to 75 days), while the feed-to-gain ratio was not significantly affected between T1, T2, and T3. 

The daily dry feed intake of growing lambs fed on dietary treatments with sprouted barley is shown in Table 3. The current results show that the daily dry feed intake of DM, OM, CP, crude fat, ash, nitrogen-free extract, fiber fractions (NDF and ADF), lignin, and mineral content (calcium, phosphorus, magnesium, sulfur, sodium) decreased linearly with increasing sprouted barley content (T2 to T5) from 1 to 75 days of the study period, compared to lambs fed with the traditional feeding system (T1) (*p* < 0.05).

For apparent digestibility for growing lambs fed on dietary treatments with sprouted barley, different levels are shown in Table 4. The digestibility of DM, OM, CP, fat, ash, NFC, NDF, ADF, and minerals such as magnesium and sulfur were linearly increased (*p* < 0.05) with sprouted barley levels (T2 to T5) compared to the lambs receiving the traditional feeding system (T1). The addition of sprouted barley at 50 to 100% with 50 to 0% traditional feed (T3 to T5; respectively) had the highest (*p* < 0.05) digestibility of CP and fat compared to the lambs receiving T1 and T2 but were not affected either linearly or quadratically. Other items’ digestibility was not affected by sprouted barley levels (*p* > 0.05), either linearly or quadratically, at 75 days of the study period.

Estimates of apparent digestibility for dietary treatments with sprouted barley at different levels in vitro are shown in Table 5. The results of the in vitro study showed that a diet of 100% sprouted barley (T5) had the quadratically lowest (*p* < 0.05) digestibility of DM, OM, and NDF compared to the traditional diet (T1) and other sprouted barley levels (T2, T3, and T4).

The pH value, Ammonia-N (NH3-N), and gas production during the in vitro digestion and fermentation for dietary treatments with different levels of sprouted barley are shown in Table 6. The culture pH value was linearly decreased (*p* < 0.05) with sprouted barley levels (T2 to T5) compared to the traditional diet (T1). NH3-N was not affected by fermentation of dietary treatments with different levels of sprouted barley (*p* > 0.05), either linearly or quadratically. The results of the in vitro study show that gas production (2 to 24 h) by digestion and fermentation of a diet of T5 (100% sprouted barley) had the highest (*p* < 0.05) compared to other treatments but was not affected either linearly or quadratically (*p* > 0.05) (Table 6; Figure 1). 

VFAs during in vitro fermentation for dietary treatments with different levels of sprouted barley are shown in Table 7. The current results indicate that acetic acid (C2) concentration was higher (*p* < 0.05) in the diet of T2 (25% sprouted barley with 75% traditional diet) compared to other treatments, except for T3 (50% sprouted barley with 50% traditional diet). There was no significant difference and it was not affected either linearly or quadratically (*p* > 0.05) between treatments. Additionally, T2 was linearly higher (*p* < 0.05) in propionic acid (C3) and total VFA compared to T1 and T4 but there was no significant difference with other treatments (T3 and T5). On the other hand, diets of 75 and 100% sprouted barley with 25 and 0% traditional diet (T4 and T5; respectively) were lower in iso-butyric acid (Iso-C4) and iso-valeric acid (Iso-Val) compared to other treatments. Butyric acid (C4) and valeric acid (Val) were not affected by sprouted barley levels (*p* > 0.05), either linearly or quadratically. Acetic acid-to-propionic acid ratio (C2:C3) had, linearly, the lowest (*p* < 0.05) in dietary treatments with sprouted barley (T2 to T5) compared to traditional diet (T1).

## 4. Discussion

The results of the chemical composition of dietary treatments showed that the content of sprouted barley DM, CP, ADF, lignin, ash, and some minerals decreased compared to traditional feeding, while NDF, fat, phosphorus, zinc, copper, TDN, and NE increased numerically. Previous studies have reported similar changes in the chemical composition of sprouted barley [2]. Fazaeli. et al. [33] and Girma and Gebremariam [7] reported that any reduction in starch content (53–67% of the dry weight of barley seeds) causes an equal reduction in OM, DM, and NFC in sprouted barley. This is in agreement with Al-Saadi and Al-Zubiadi [17], who showed that crude protein content was higher in sprouted barley compared to untreated grain, which could be due to the use of carbohydrates to provide energy to the seeds by respiration. NDF and ADF were also increased, but NFC decreased in sprouted barley compared to barley grain on a DM basis [7].

In the current study, used sprouted barley at 25–100% compared to 75–0% traditional diet had the lowest FBW, while BWG and ADG was lower only in lambs fed 100% sprouted barley. In addition, our results showed that the average daily intake and feed-to-gain ratio were increased with sprouted barley compared to lambs fed with the traditional diet (T1). These results agree with the reports of Muhammad et al. [34] that feeding sprouted barley with a traditional diet may have a negative effect on growth performance compared to a concentrate diet. This could be due to the lower daily dry feed intake which allows animals to become full without meeting the nutrient requirements for optimal performance. Morales et al. [35] reported that sprouted barley contains a lower DM, so animals may not be able to meet their dry feed intake requirements, and this may have a negative effect on growth. In addition, the lower DM in sprouted barley resulted in a lower OM value as it constitutes a major part of the DM. The OM, especially starch, may be consumed to support metabolism and energy requirements during sprout [36]. A study by Saidi and Jamal [4] showed that sprouted barley did not affect the bodyweight of ewes. However, Ata [1] found that lambs fed 62% sprouted barley as part of the total mixed ratio (TMR) had a higher total weight gain, average daily gain, average daily intake, and better feed-to-gain ratio than TMR (control diet). In another study, it was shown that goats fed sprouted barley had a higher total intake of DM and weight gain compared to concentrate diets [37]. Moreover, Fayed [3] found that feeding sprouted barley on rice straw had a positive effect on the growth performance of lambs. 

The current results show that the apparent digestibility of DM, OM, CP, fat, ash, NFC, NDF, ADF, and minerals such as Mg and sulfur increased linearly with the level of sprouted barley compared to lambs fed traditional diets may be due to the easy degradation of sprouted barley in the rumen. These results agree with those of Al-Saadi and Al-Zubiadi [17], who reported that digestibility of DM, OM, CP, and fat was higher in sprouted barley than cereal barley. Similarly, Morgan et.al. [36] indicated that the digestibility of OM and DM was higher with the addition of sprouted barely. Nutrient digestibility was increased in lambs fed 33, 66, and 100% sprouted barley compared to traditional feeding [38]. In contrast, Dung et al. [13] reported that sprouted barely did not affect nutrient digestibility in ruminants due to total loss DM, while the lowest rumen pH value and the highest VFAs. Fayed [3] showed that apparent DM, OM, CP, and fat digestibility were higher when sprouted barley was combined with rice straw in the diet of lambs. On the other hand, the estimation of apparent digestibility in vitro for the different feed treatments with sprouted barley showed that the diet with 100% sprouted barley had the lowest digestibility of DM, OM and NDF compared to the other diets. These results agree with Fazaeli et al. [6] who reported that nutrient digestibility of sprouted barley was lower than that of pure cereals in vitro through rumen fluid in glass syringes. The high nutrient digestibility in vivo could be due to the high content of leaves and roots, which are easily digested and hydrolyzed by the enzymes of the microflora in the rumen of the lambs, and enzymatic digestion in the lytic vacuoles of the plant cells, which may lead to differences in the digestibility of the feed in vivo and in vitro, which was digested as DM. Ikram et al. [39] showed that many biochemical alterations occur during germination, affecting digestibility due to enzymes that split up carbohydrates and proteins into basic compounds in barley seed. 

Our findings that sprouted barley lowers the pH of the culture in vitro. These results agree with the in vivo study of Al-Saadi and Al-Zubiadi [17] who found that lambs fed the sprouted barley had lower rumen fluid pH than grain barley. The low pH at different levels of sprouted barley could also indicate a change in the rumen ecosystem, such as microflora activity or production of VFAs due to the utilization of sprouted barley. However, the concentration of VFAs was higher in lambs fed fresh sprouted barley [40]. Our results are consistent with the fact that 25% sprouted barley with 75% traditional diet (T2) had higher propionic acid and total VFA concentration and high acetic acid concentration in the diet of T2 compared to T1 and T4, which may indicate that sprouted barley enhances carbohydrate fermentation in the rumen and absorption of VFAs [3,9]. In addition, our results suggest that the levels of VFAs during in vitro fermentation for the sprouted barley feed treatments are within the ranges found in continuous culture fermentation studies [41,42]. The sprouted barley forage treatments had numerically higher NFC values compared to the traditional forages, which could be due to the higher concentration of total VFA and propionic acid. The acetic acid concentration was higher in the diet of T2, which may be due to the high NDF content that contributed to higher acetate concentration compared to the traditional diet (T1).

## 5. Conclusions

From the results of the current study, it can be concluded that the addition of sprouted barley with traditional diet improved the nutritional value, the digestibility of diet composition, and fermentation characteristics. On the other hand, the addition of sprouted barley has a negative effect on growth performance and lower daily dry feed intake. So, it may be a good strategy if further studies are conducted to increase nutrient requirements for optimal growth performance. 

## Figures and Tables

**Figure 1 animals-12-01206-f001:**
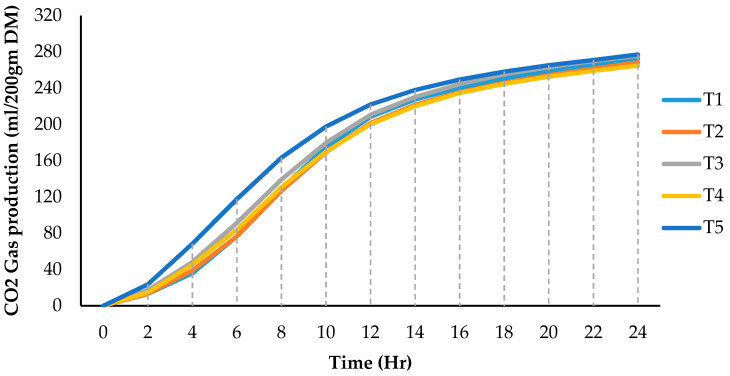
Cumulative CO2 gas production during the in vitro digestion for dietary treatments from 2 to 24 h.

**Table 1 animals-12-01206-t001:** Chemical composition of the diets fed growing lambs on a DM basis ^1^.

Items ^2^	Treatments ^3^				
	T1	T2	T3	T4	T5
DM, %	95.5	76.7	57.8	38.9	20.1
CP, %	15.01	14.64	14.43	14.23	13.86
NDF, %	34.21	35.00	30.26	31.05	36.90
ADF, %	19.78	18.25	16.03	17.57	16.82
Lignin, %	5.78	5.41	4.96	4.12	2.10
NFC, %	43.86	43.60	48.77	48.34	43.50
Fat, %	1.68	1.81	1.90	2.01	2.69
Ash, %	5.24	4.95	4.64	4.37	3.05
Calcium, %	0.65	0.65	0.53	0.45	0.19
Phosphorus, %	0.23	0.25	0.27	0.30	0.39
Magnesium, %	0.18	0.19	0.19	0.18	0.14
Potassium, %	1.47	0.40	1.20	1.04	0.51
Sulfur, %	0.22	0.23	0.22	0.21	0.18
Sodium, %	0.20	0.19	0.20	0.18	0.15
Zinc, ppm	33.00	37.00	39.00	41.00	70.00
Copper, ppm	5.00	7.00	6.00	5.00	6.00
TDN	80.00	81.80	84.30	82.50	83.40
NE, Mcal/lb	0.84	0.86	0.88	0.86	0.87

^1^ The chemical composition analysis was performed in triplicate; ^2^ DM = Dry matter; CP = Crude protein; NDF = Neutral detergent fiber; ADF = Acid detergent fiber; NFC = Non-fibrous carbohydrates [NFC = 100% − (CP + total fiber + fat + ash)]; TDN = Total digestible nutrients [TDN = dig CP + dig fiber + dig NFC + (2.25 × dig fat)]; NE = Net energy. ^3^ Treatments, T1: 100% traditional diet (Barley 70: Alfalfa hay 30); T2: 75% traditional diet with 25% sprouted barley; T3: 50% traditional diet with 50% sprouted barley; T4: 25% traditional diet with 75% sprouted barley and T5: 100% sprouted barley.

**Table 2 animals-12-01206-t002:** Growth performance for growing lambs fed on dietary treatments with different levels of sprouted barley from 1 to 75 days.

Items ^2^	Treatments ^1^					SEM ^3^	*p*-Value		
	T1	T2	T3	T4	T5		Treat.	Linear	Quadratic
IBW (kg)	27.8	27.6	27.6	27.8	28.10	0.91	0.986	0.903	0.597
FBW (kg)	41.0 ^a^	38.2 ^b^	38.1 ^b^	38.2 ^b^	31.0 ^c^	1.88	0.041	0.048	0.407
BWG (kg)	13.18 ^a^	10.60 ^a^	10.50 ^a^	10.35 ^a^	2.90 ^b^	1.46	0.009	0.014	0.185
ADG (g/d)	175.8 ^a^	140.6 ^a^	143.6 ^a^	137.8 ^a^	39.0 ^b^	19.5	0.009	0.014	0.185
ADI (g/d)	1117 ^d^	1338 ^c^	1587 ^b^	1874 ^a^	1672 ^b^	63.1	<0.0001	<0.0001	0.013
FI: WG (g:g)	7.2 ^c^	10.2 ^c^	11.8 ^bc^	17.6 ^b^	43.8 ^a^	2.15	<0.0001	<0.0001	<0.0001
RG %	41.5 ^a^	31.6 ^a^	32.3 ^a^	35.8 ^a^	10.0 ^b^	3.29	<0.0001	<0.0001	0.156

^a–d^ Means values within rows for each item with clarification of the significant difference in the form of superscripts (*p* < 0.05). ^1^ Treatments, T1: 100 % traditional diet (Barley 70: Alfalfa hay 30); T2: 75% traditional diet with 25% sprouted barley; T3: 50% traditional diet with 50% sprouted barley; T4: 25% traditional diet with 75% sprouted barley and T5: 100% sprouted barley. ^2^ IBW = Initial body weight; FBW = Final body weight; BWG = Weight gain; ADG = Average daily gain; ADI = Average daily intake (g/d); FI: WG = Feed-to-gain ratio; RG = Relative growth. ^3^ SEM = Standard error of means for treatments effect.

**Table 3 animals-12-01206-t003:** Daily dry feed intake (g/day) for growing lambs fed on dietary treatments with different levels of sprouted barley from 1 to 75 days.

Items ^2^	Treatments ^1^					SEM ^3^	*p*-Value		
	T1	T2	T3	T4	T5		Treat.	Linear	Quadratic
DM	1465 ^a^	1314 ^a^	953 ^b^	777 ^b^	232 ^c^	97.7	<0.0001	<0.0001	0.112
OM	1388 ^a^	1249 ^a^	909 ^b^	743 ^b^	224 ^c^	93.1	<0.0001	<0.0001	0.107
CP	219.8 ^a^	192.4 ^a^	137.5 ^b^	110.6 ^b^	32.2 ^c^	14.1	<0.0001	<0.0001	0.175
NDF	501.1 ^a^	460.1 ^a^	288.4 ^b^	241.4 ^b^	85.6 ^c^	31.4	<0.0001	<0.0001	0.404
ADF	289.7 ^a^	239.9 ^b^	152.8 ^c^	136.6 ^c^	39.0 ^d^	16.5	<0.0001	<0.0001	0.713
Lignin	84.6 ^a^	71.1 ^a^	47.2 ^b^	32.0 ^c^	4.8 ^d^	4.91	<0.0001	<0.0001	0.314
NFC	642.5 ^a^	573.1 ^ab^	464.8 ^bc^	375.9 ^c^	100.9 ^d^	45.8	<0.0001	<0.0001	0.033
Fat	24.6 ^a^	23.7 ^a^	18.1 ^b^	15.6 ^b^	6.2 ^c^	1.84	<0.0001	0.0008	0.056
Ash	76.7 ^a^	65.0 ^a^	44.2 ^b^	33.9 ^b^	7.0 ^c^	4.61	<0.0001	<0.0001	0.263
Calcium	9.52 ^a^	8.54 ^a^	5.05 ^b^	3.50 ^b^	0.44 ^c^	0.55	<0.0001	<0.0001	0.317
Phosphorus	3.36 ^a^	3.28 ^a^	2.57 ^ab^	2.33 ^b^	0.90 ^c^	0.26	<0.0001	0.001	0.035
Magnesium	2.63 ^a^	2.49 ^a^	1.81 ^b^	1.40 ^b^	0.32 ^c^	0.18	<0.0001	<0.0001	0.029
Sulfur	3.22 ^a^	3.02 ^a^	2.09 ^b^	1.63 ^b^	0.41 ^c^	0.21	<0.0001	<0.0001	0.067
Sodium	2.92 ^a^	2.49 ^a^	1.90 ^b^	1.40 ^b^	0.34 ^c^	0.19	<0.0001	<0.0001	0.107

^a–d^ Means values within rows for each item with clarification of the significant difference in the form of superscripts (*p* < 0.05). ^1^ Treatments, T1: 100 % traditional diet (Barley 70: Alfalfa hay 30); T2: 75% traditional diet with 25% sprouted barley; T3: 50% traditional diet with 50% sprouted barley; T4: 25% traditional diet with 75% sprouted barley and T5: 100% sprouted barley. ^2^ DM = Dry matter; OM = Organic matter; CP = Crude protein; NDF = Neutral detergent fiber; ADF = Acid detergent fiber; NFC = Non-fibrous carbohydrates. ^3^ SEM = Standard error of means for treatments effect.

**Table 4 animals-12-01206-t004:** Apparent digestibility (%) for growing lambs fed on dietary treatments with different levels of sprouted barley at 75 days.

Items ^2^	Treatments ^1^					SEM ^3^	*p*-Value		
	T1	T2	T3	T4	T5		Treat.	Linear	Quadratic
DM, %	81.80 ^b^	81.59 ^b^	87.33 ^a^	88.15 ^a^	90.89 ^a^	1.69	0.004	0.015	0.882
OM, %	82.73 ^b^	82.47 ^b^	88.34 ^a^	88.88 ^a^	91.46 ^a^	1.63	0.004	0.014	0.955
CP, %	72.41 ^b^	69.61 ^b^	79.19 ^a^	83.46 ^a^	86.77 ^a^	3.46	0.015	0.077	0.601
NDF, %	65.31 ^c^	67.51 ^bc^	74.49 ^bc^	78.40 ^ab^	86.71 ^a^	3.84	0.009	0.017	0.511
ADF, %	67.96 ^bc^	66.38 ^c^	73.03 ^bc^	74.68 ^b^	82.25 ^a^	2.46	0.003	0.042	0.162
Lignin, %	72.40	74.42	80.17	77.29	73.59	3.33	0.494	0.302	0.112
NFC, %	99.54	98.79	99.26	98.51	99.06	0.63	0.812	0.388	0.560
Fat, %	65.21 ^c^	61.04 ^c^	73.92 ^b^	81.05 ^ab^	89.53 ^a^	5.95	0.025	0.114	0.379
Ash, %	64.93	64.56	66.71	72.41	72.67	3.80	0.392	0.344	0.728
Calcium, %	40.58	42.03	43.86	59.74	36.23	7.57	0.279	0.572	0.246
Phosphorus, %	47.85	57.03	55.34	50.80	66.78	5.35	0.175	0.128	0.614
Magnesium, %	39.91 ^b^	41.84 ^b^	58.50 ^a^	68.08 ^a^	67.76 ^a^	4.65	0.0008	0.002	0.527
Sulfur, %	65.83 ^b^	65.90 ^b^	76.15 ^ab^	78.54 ^a^	79.10 ^a^	3.27	0.017	0.025	0.582
Sodium, %	72.97	75.32	82.08	83.65	75.68	5.22	0.547	0.303	0.186

^a–c^ Means values within rows for each item with clarification of the significant difference in the form of superscripts (*p* < 0.05). ^1^ Treatments, T1: 100 % traditional diet (Barley 70: Alfalfa hay 30); T2: 75% traditional diet with 25% sprouted barley; T3: 50% traditional diet with 50% sprouted barley; T4: 25% traditional diet with 75% sprouted barley and T5: 100% sprouted barley. ^2^ DM = Dry matter; OM = Organic matter; CP = Crude protein; NDF = Neutral detergent fiber; ADF = Acid detergent fiber; NFC = Non-fibrous carbohydrates. ^3^ SEM = Standard error of means for treatments effect.

**Table 5 animals-12-01206-t005:** Apparent digestibility for dietary treatments with different levels of sprouted barley in vitro.

Items ^2^	Treatments ^1^					SEM ^3^	*p*-Value		
	T1	T2	T3	T4	T5		Treat.	Linear	Quadratic
DM, %	79.4 ^a^	79.9 ^a^	82.7 ^a^	82.5 ^a^	72.6 ^b^	1.16	0.0007	0.979	0.001
OM, %	80.0 ^a^	80.4 ^a^	82.8 ^a^	82.8 ^a^	74.3 ^b^	0.98	0.0007	0.924	0.001
NDF, %	53.3 ^a^	55.4 ^a^	53.6 ^a^	54.1 ^a^	46.4 ^b^	1.74	0.030	0.634	0.020

^a,b^ Means values within rows for each item with clarification of the significant difference in the form of superscripts (*p* < 0.05). ^1^ Treatments, T1: 100 % traditional diet (Barley 70: Alfalfa hay 30); T2: 75% traditional diet with 25% sprouted barley; T3: 50% traditional diet with 50% sprouted barley; T4: 25% traditional diet with 75% sprouted barley and T5: 100% sprouted barley. ^2^ DM = dry matter; OM = Organic matter; NDF = Neutral detergent fiber. ^3^ SEM = Standard error of means for treatments effect.

**Table 6 animals-12-01206-t006:** The pH value, Ammonia-N and CO_2_ gas production during the in vitro digestion and fermentation for dietary treatments with different levels of sprouted barley.

Items ^2^	Treatments ^1^					SEM ^3^	*p*-Value		
	T1	T2	T3	T4	T5		Treat.	Linear	Quadratic
Culture pH	6.13 ^a^	6.02 ^b^	6.02 ^b^	6.01 ^b^	5.92 ^b^	0.03	0.010	0.002	0.669
NH3-N, mM	4.12	4.54	3.99	4.28	4.85	0.68	0.902	0.707	0.643
TGP, mL	179.3 ^b^	176.1 ^b^	185.4 ^b^	176.7 ^b^	196.0 ^a^	3.13	0.005	0.255	0.108

^a,b^ Means values within rows for each item with clarification of the significant difference in the form of superscripts (*p* < 0.05). ^1^ Treatments, T1: 100 % traditional diet (Barley 70: Alfalfa hay 30); T2: 75% traditional diet with 25% sprouted barley; T3: 50% traditional diet with 50% sprouted barley; T4: 25% traditional diet with 75% sprouted barley and T5: 100% sprouted barley. ^2^ NH3 = Ammonia-N; TGP = total CO_2_ gas production. ^3^ SEM = Standard error of means for treatments effect.

**Table 7 animals-12-01206-t007:** Volatile fatty acids (VFA) during the in vitro fermentation for dietary treatments with sprouted barley different levels.

Items ^2^	Treatments ^1^					SEM ^3^	*p*-Value		
	T1	T2	T3	T4	T5		Treat.	Linear	Quadratic
C2, mM	30.5 ^b^	33.8 ^a^	32.2 ^ab^	29.3 ^b^	30.8 ^b^	0.90	0.042	0.331	0.301
C3, mM	12.1 ^c^	15.0 ^a^	13.9 ^ab^	12.7 ^bc^	14.1 ^ab^	0.53	0.019	0.011	0.289
Iso-C4, mM	0.46 ^a^	0.49 ^a^	0.45 ^ab^	0.37 ^c^	0.39 ^bc^	0.02	0.010	0.170	0.472
C4, mM	9.3	10.5	10.1	9.1	10.5	0.64	0.423	0.294	0.918
Iso-Val, mM	0.91 ^a^	0.90 ^a^	0.86 ^a^	0.74 ^b^	0.82 ^ab^	0.03	0.028	0.072	0.598
Val, mM	0.95	1.07	1.03	0.91	1.08	0.06	0.351	0.358	0.936
C2:C3	2.51 ^a^	2.24 ^b^	2.31 ^b^	2.30 ^b^	2.18 ^b^	0.05	0.016	0.002	0.367
Total VFA, mM	54.2 ^b^	61.9 ^a^	58.6 ^ab^	53.2 ^b^	57.8 ^ab^	1.93	0.050	0.123	0.409

^a–c^ Means values within rows for each item with clarification of the significant difference in the form of superscripts (*p* < 0.05). ^1^ Treatments, T1: 100 % traditional diet (Barley 70: Alfalfa hay 30); T2: 75% traditional diet with 25% sprouted barley; T3: 50% traditional diet with 50% sprouted barley; T4: 25% traditional diet with 75% sprouted barley and T5: 100% sprouted barley. ^2^ C2 = acetic acid; C3 = propionic acid; C4 = butyric acid; Val = valeric acid; Total VFA = total volatile fatty acids. ^3^ SEM = Standard error of means for treatments effect.

## Data Availability

All data sets collected and analyzed during the current study are available from the corresponding author on fair request.

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
