# Peer review of "Evaluation of the Nutritive Value and Digestibility of Sprouted Barley as Feed for Growing Lambs: In Vivo and In Vitro Studies"

_animals, 2022, doi:10.3390/ani12091206_

Round 1
Reviewer 1 Report
Dear editor
Pls see the attached file.
REGARDS

Author Response
thanks for your valuable comments and please find our responses below:
Q1: Lines 21-39. The ABSTRACT concisely explained the study (purpose, material and methods, results, conclusion).
Authors’ Response: Thank you for pointing out that. We revised as required.
Q2: Lines 21- 39. Abbreviations should be introduced in the text when first used.
Authors’ Response: Done as requested
Q3: Lines 58-62. Old references
Authors’ Response: Thanks for your feedback. Done as requested.
Q4: Lines 74. SEVIM, B. AYAÅžAN T., ÜLGER, Ä°., ERGÜL Åž., AYKANAT S., COÅžKUN MA., 2017. Determination of nutritive values of different malt barley varieties using in vitro gas production technique.Turkish Journal of Agriculture - Food Science and Technology, 5(10): 1216-1220. DOI: https://doi.org/10.24925/turjaf.v5i10.1216-1220.1386
Thank you for your comments. Through the reference referred to, the nutritional value of the barley grain varieties was determined but our study is to determine the nutritional and digestibility value of sprouted barley compared with traditional feed to verify the hypothesis and answer the question: Can sprouted barley be replaced as part or complete in the feeding of growing lambs? Through the results obtained in the current study. Please if there is another opinion, guide us to correct it.
Q5: Lines 89. The design of the MATERIAL and METHODS appropriate and relevant to the purpose of the study.
Authors’ Response: Thanks for your feedback. Done as requested.
Q6: Lines 126. what are initial body weights? ( )???
Authors’ Response: Done as requested
Q7: Lines 138-142. Give relevant reference
Authors’ Response: Done as requested.
Q8: Lines 209. The results discussed with previous findings were scientifically compared.
Authors’ Response: Thank you for pointing out that.
Q9: Lines Table 3, 4, 5, 6, 7. Why bold?
Thank you for your notes. There was an error while typing and it has been modified.
Q10: Lines 338. More discussion is required. More references
Authors response: We revised as required with updated references.
Q11: Lines 351. Thorough discussion of the findings is required wit support of literature especially current sources (5 years old i.e. from 2015 to date).
Authors response: Thank you for your comments. We revised as required.
Q12: Lines 432-515. Refer. Not unifomity
Thank you for your notes. It has been modified in references (referred).

Reviewer 2 Report
Evaluation of nutritive value and digestibility of sprouted bar-2 ley for growing lambs feeding: in vivo and in vitro studies
The authors clearly justify the present work and present a nice piece of work with supporting information from both, in vivo and in vitro studies.
From my point of view, I found the work well planned and executed. Only minor questions to be solved (see file).
However, I found the text sometimes difficult to read (in some sections) and I believe it would be greatly enhanced if an English language editor with experience in animal science revise the whole text.
Regards

Author Response
thanks for your valuable comments and please find our responses below:
Q1: Line 11. Writing needs to be revised for clarity by a professional service.
Authors’ Response: Thanks for your feedback. We have improved this section in manuscript.
Q2: Line12. some modifications
Authors’ Response: Done as requested in line 12.
Q3: Line 32. p < or >?
Authors’ Response: Thanks so much for your notification. We have revised in line 35.
Q4: Line 39. Several sentences need improvement.
Authors’ Response: Thanks for your feedback. We have improved in lines 43- 83 (Introduction).
Q5: Line 50. forage?
Authors’ Response: Done as requested. Line 54.
Q6: Line 99. why cut and provided whole to the animals?
Authors’ Response: Thanks so much for your question. The sprouted barley is cut so that it is easier for the animals to eat and so that the animals consume it completely without picking only the leaves.
Q7: Line 128. amount? chemical composition of diet? was any nutrient requirement standard used as reference?.
Authors’ Response: Thanks so much for your noted. Nutrient requirements relied on traditional and common diets in which animals are fattened based on the ratio that was conducted in previous studies, which is 70% barley grain + 30% alfalfa hay.
The composition of diets and Analysis are shown in Table 1.
During the study period, all animals were provided ad libitum access to feed with 5% additional feed daily to reach refusal.

Reviewer 3 Report
Line 25: please reword the phrase “DM and non-fibrous carbohydrates were measurements”. It sounds incomplete.
Line 31: Please reword the phase “In the vivo study” to “In the in vivo study”.
Line 33: Delete the word “study”.
Line: Change the word “presented” to “offered”
Line 102 “Diet samplings” or “Sampling of diets”
Line 120 “Animal housing”
Line 122: delete the word “trustworthy”
Line 178: The first sentence in this paragraph seems incomplete. Please reword.
Line 180: delete the word “placed”
Line 184: Not need to indicate the number of carbons
Line 214: Please, correct the title “.fed to growing lambs.”
In Table 1, for the percent inclusion of major nutrient, only use 3 digits or 1 decimal.
Line 122: delete the word “growing”
Line 132: “feed to gain”
Line 243: delete the word “on”
Line 251, 270 and other Footnote Tables: “Mean values…” or “Means…”. This sentence is not clear, please reword. Do you simply want to say: “Means with different superscript are statistically different (P-value)”?
Line 337: “In agreement with the present experiment…”
Line 343 “…the use…”
Line 383: “Our findings indicate”
Line 387: “microbiota”
Line 401: delete the word “be” and “concluded” in present tense.
Author Response
Dear Reviewer
thanks for your valuable comments and please find our responses below:
The manuscript presents the study performed on lambs fed different level of sprouted barley. The process by which SB was received is also included into the manuscript. It is very interesting study. From the point of view of proper fermentation in rumen and the effect on the performance this study is important.
Authors’ Response: Thank you for your comment.
Q1: Line 23. the abbreviation should be explained.
Authors’ Response: Thanks so much for your notification. We have revised in line 23.
Q2: Line 24, 25. abbr.
Authors’ Response: Done as requested in lines 24-25 .
Q3: Line 26. were measured
Authors’ Response: Done as requested in line 27.
Q4: Line 27. without abbr.
Authors’ Response: Done as requested in line 28.
Q5: in general, abstract is hard to read due to many abbreviation
Authors’ Response: Thanks for your feedback. We have modified many abbreviations in the abstract.
Q6: Line 49. why “water” by capital letter?
Authors’ Response: Thank you for your notes. There was an error while typing and it has been modified in line 53.
Q7: Line 65. using for the first time the abbr. it should be explained.
Authors’ Response: Thank you for your notes. Done as requested in line 69.
Q8: On the other hand, the introduction is very well written,
Authors’ Response: Thanks for your feedback.
Q9: Line 109. the abbre. Should be explained
Authors’ Response: Done as requested in lines 113-114 .
Q10: Line 114. EE?
Authors’ Response: Thanks for your feedback. We have deleted this term and rewritten it correctly in line 118.
Q11: Line 115. writing elements should be uniformed
Authors’ Response: Thank you for your notes. Done as requested in line 120.
Q12: Line 141. it is better to weight every day for the determination of daily gain or average wight gain, but using only to values like initial and final it is also possible, although it is harder to follow of changes in body weight, because it is not known if the gain was equal in the same periods
Authors’ Response: Thank you for this constructive opinion. Performance, such as daily gain or average weight gain and feed intake, were the same during the study period.
Therefore, we believe that the average performance during the study is sufficient because the study aims to verify the nutritional value and digestibility and then to investigate the effects on performance.
Please, if there are any opinions, guide us to it.
Q13: Line 210-211. unify
Authors’ Response: Done as requested in lines 214-217.
Q14: Table 1 – all components should be added to present proper calculation
Authors’ Response: Done as requested.
Q15: Table 3 – Sulfer?
Authors’ Response: Thank you to correct us for this term. We have corrected this term in all tables.
Q16: Line 245. unify along the manuscript, after using abbr. for the first time, they should be use in the text
Authors’ Response: Done as requested in using abbr after the first time to unify along the manuscript in lines 251-252.
Q17: Line 246, 333. unify
Authors’ Response: Done as requested in lines 253-254, 341.
Q18: Line 292-295. is it the same ammonia or not
Authors’ Response: Thanks so much for your question. no, NH3 and NH3-N are different expressions of the chemical forms of ammonia. The NH3-N form uses the molecular weight of only the nitrogen atoms. Also, ammonia is an inorganic compound having the chemical formula NH3 whereas ammonia nitrogen is a measure of the amount of ammonia in a sample.
Q19: Table 6 what is “PH”
Authors’ Response: The term pH (not PH) is a quantitative measure of the acidity or basicity of any solution.
Q20: Table 6 - why the component of gas produced was not determined. Especially NH4 and CO2
Authors’ Response: Done as requested.
Q21: “we can be concluded”?
Authors’ Response: Done as requested in line 410.
The manuscript has been completely revised, with some language changes made and improved from our point of view, on the other hand, I appreciate your efforts in your valuable comments and question, which gave me the opportunity to improve the throughout manuscript.
Please if there are any opinions, guide us to correct it.
satisfactory to you
Thanks so much for your efforts. Your feedbacks are very valuable and will improve my research skills and biological insight on my future studies.

Reviewer 4 Report
Comments to the Authors of manuscript number: animals-1712373 entitled “Evaluation of nutritive value and digestibility of sprouted barley for growing lambs feeding: in vivo and in vitro studies”.
The manuscript presents the study performed on lambs fed different level of sprouted barley. The process by which SB was received is also included into the manuscript. It is very interesting study. From the point of view of proper fermentation in rumen and the effect on the performance this study is important.
- L 23 – the abbreviation should be explained.
- L 24, 25 – abbr.
- L 26 – were measured
- L 27 – without abbr.
- in general, abstract is hard to read due to many abbreviation
- L 49 – why “water” by capital letter?
- L 65 – using for the first time the abbr. it should be explained.
- On the other hand, the introduction is very well written,
- L 109 – the abbre. Should be explained
- L 114- EE?
- L 115 – writing elements should be uniformed
- L 141- it is better to weight every day for the determination of daily gain or average wight gain, but using only to values like initial and final it is also possible, although it is harder to follow of changes in body weight, because it is not known if the gain was equal in the same periods
- 210-211 -unify
- Table 1 – all components should be added to present proper calculation
- Table 3 – Sulfer?
- L 245 – unify along the manuscript, after using abbr. for the first time, they should be use in the text
- L 246, 333 - unify
- L 292-295 is it the same ammonia or not
- Table 6 what is “PH”
- Table 6 - why the component of gas produced was not determined. Especially NH4 and CO2
- “we can be concluded”?
Author Response
Dear Reviewer
Thanks for your valuable comments and please find our responses below:
The manuscript presents the study performed on lambs fed different level of sprouted barley. The process by which SB was received is also included into the manuscript. It is very interesting study. From the point of view of proper fermentation in rumen and the effect on the performance this study is important.
Authors’ Response: Thank you for your comment.
Q1: Line 23. the abbreviation should be explained.
Authors’ Response: Thanks so much for your notification. We have revised in line 23.
Q2: Line 24, 25. abbr.
Authors’ Response: Done as requested in lines 24-25 .
Q3: Line 26. were measured
Authors’ Response: Done as requested in line 27.
Q4: Line 27. without abbr.
Authors’ Response: Done as requested in line 28.
Q5: in general, abstract is hard to read due to many abbreviation
Authors’ Response: Thanks for your feedback. We have modified many abbreviations in the abstract.
Q6: Line 49. why “water” by capital letter?
Authors’ Response: Thank you for your notes. There was an error while typing and it has been modified in line 53.
Q7: Line 65. using for the first time the abbr. it should be explained.
Authors’ Response: Thank you for your notes. Done as requested in line 69.
Q8: On the other hand, the introduction is very well written,
Authors’ Response: Thanks for your feedback.
Q9: Line 109. the abbre. Should be explained
Authors’ Response: Done as requested in lines 113-114 .
Q10: Line 114. EE?
Authors’ Response: Thanks for your feedback. We have deleted this term and rewritten it correctly in line 118.
Q11: Line 115. writing elements should be uniformed
Authors’ Response: Thank you for your notes. Done as requested in line 120.
Q12: Line 141. it is better to weight every day for the determination of daily gain or average wight gain, but using only to values like initial and final it is also possible, although it is harder to follow of changes in body weight, because it is not known if the gain was equal in the same periods
Authors’ Response: Thank you for this constructive opinion. Performance, such as daily gain or average weight gain and feed intake, were the same during the study period.
Therefore, we believe that the average performance during the study is sufficient because the study aims to verify the nutritional value and digestibility and then to investigate the effects on performance.
Please, if there are any opinions, guide us to it.
Q13: Line 210-211. unify
Authors’ Response: Done as requested in lines 214-217.
Q14: Table 1 – all components should be added to present proper calculation
Authors’ Response: Done as requested.
Q15: Table 3 – Sulfer?
Authors’ Response: Thank you to correct us for this term. We have corrected this term in all tables.
Q16: Line 245. unify along the manuscript, after using abbr. for the first time, they should be use in the text
Authors’ Response: Done as requested in using abbr after the first time to unify along the manuscript in lines 251-252.
Q17: Line 246, 333. unify
Authors’ Response: Done as requested in lines 253-254, 341.
Q18: Line 292-295. is it the same ammonia or not
Authors’ Response: Thanks so much for your question. no, NH3 and NH3-N are different expressions of the chemical forms of ammonia. The NH3-N form uses the molecular weight of only the nitrogen atoms. Also, ammonia is an inorganic compound having the chemical formula NH3 whereas ammonia nitrogen is a measure of the amount of ammonia in a sample.
Q19: Table 6 what is “PH”
Authors’ Response: The term pH (not PH) is a quantitative measure of the acidity or basicity of any solution.
Q20: Table 6 - why the component of gas produced was not determined. Especially NH4 and CO2
Authors’ Response: Done as requested.
Q21: “we can be concluded”?
Authors’ Response: Done as requested in line 410.
The manuscript has been completely revised, with some language changes made and improved from our point of view, on the other hand, I appreciate your efforts in your valuable comments and question, which gave me the opportunity to improve the throughout manuscript.
Please if there are any opinions, guide us to correct it.
satisfactory to you
Thanks so much for your efforts. Your feedbacks are very valuable and will improve my research skills and biological insight on my future studies.
